# From Barlow Twins to Triplet Training: Differentiating Dementia with Limited Data

**Yitong Li**[*1,2], **Tom Nuno Wolf**[*1,2], **Sebastian Pölsterl**[1], **Igor Yakushev**[3],
**Dennis M. Hedderich**[4], **Christian Wachinger**[1,2]    YI_TONG.LI@TUM.DE

[1] *Laboratory for Artificial Intelligence in Medical Imaging, Department of Radiology, Technical University of Munich (TUM), Germany*

[2] *Munich Center for Machine Learning (MCML), Germany*

[3] *Department of Nuclear Medicine, Klinikum rechts der Isar, TUM, Germany*

[4] *Department of Neuroradiology, Klinikum rechts der Isar, TUM, Germany*

**Editors:** Accepted for publication at MIDL 2024

## Abstract

Differential diagnosis of dementia is challenging due to overlapping symptoms, with structural magnetic resonance imaging (MRI) being the primary method for diagnosis. Despite the clinical value of computer-aided differential diagnosis, research has been limited, mainly due to the absence of public datasets that contain diverse types of dementia. This leaves researchers with small in-house datasets that are insufficient for training deep neural networks (DNNs). Self-supervised learning shows promise for utilizing unlabeled MRI scans in training, but small batch sizes for volumetric brain scans make its application challenging. To address these issues, we propose *Triplet Training* for differential diagnosis with limited target data. It consists of three key stages: (i) self-supervised pre-training on unlabeled data with Barlow Twins, (ii) self-distillation on task-related data, and (iii) fine-tuning on the target dataset. Our approach significantly outperforms traditional training strategies, achieving a balanced accuracy of 75.6%. We further provide insights into the training process by visualizing changes in the latent space after each step. Finally, we validate the robustness of Triplet Training in terms of its individual components in a comprehensive ablation study. Our code is available at https://github.com/ai-med/TripletTraining.

**Keywords:** differential diagnosis, dementia, transfer learning, limited data.

## 1. Introduction

The number of patients suffering from dementia is expected to increase to 152.8 million by 2050 (Nichols et al., 2022), with Alzheimer's Disease (AD) accounting for 60-80% of affected patients. Frontotemporal dementia (FTD) is the second most common type of dementia in the younger-elderly population (aged < 65 years) (Young et al., 2018). Accurately diagnosing different dementia types is challenging as symptoms overlap, but is crucial for patient management, therapy, and prognosis. In the clinical routine, differential diagnosis incorporates structural magnetic resonance imaging (sMRI) to evaluate distinct atrophy patterns. Despite the clinical importance of differential diagnosis, there is limited research in computer-aided diagnosis for this task compared to classifying AD and cognitively normal (CN) subjects, largely rooted in the lack of related public MRI datasets. Accessing in-house

---

[*] Contributed equally

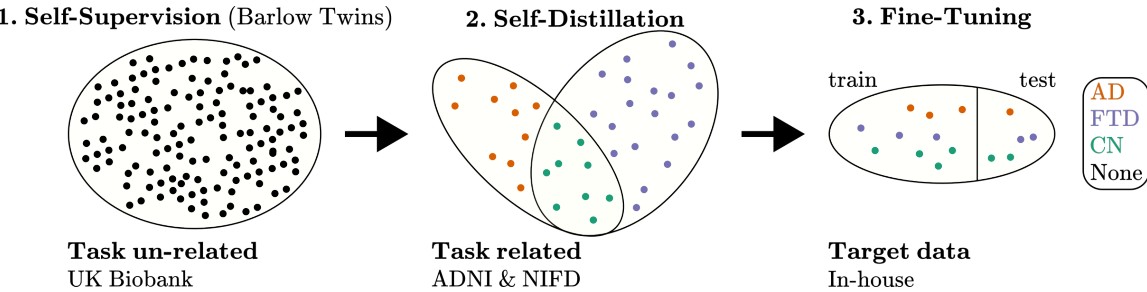

Figure 1: Triplet Training for differential diagnosis of dementia: 1) task un-related data is invoked with self-supervision, 2) self-distillation on task-related data, 3) the network is fine-tuned on the training part of the target dataset and evaluated on the test part.

data from hospitals is an alternative; however, even if available, such data is typically too small to train DNNs successfully.

At the same time, public datasets exist that focus on single types of dementia. For AD, the Alzheimer's disease neuroimaging initiative (ADNI, adni.loni.usc.edu) provides an extensive resource (Jack et al., 2008). Similarly, the initiative on Neuroimaging in Frontotemporal Dementia (NIFD, 4rtni-ftldni.ini.usc.edu) collected data for FTD. As a result, previous research on the differential diagnosis of AD and FTD combined the two datasets (Ma et al., 2020; Hu et al., 2021; Nguyen et al., 2022). An inherent limitation of such a combination is the confounding of dataset and diagnosis, potentially yielding shortcut learning that differentiates datasets instead of diagnosis (Geirhos et al., 2020). While the evaluation of such a merged dataset easily leads to inflated estimates of classification accuracy, it can instead provide a valuable resource in the training process.

Population imaging studies, e.g., UK Biobank (Miller et al., 2016), establish an even larger resource of MRI data for training, but they do not contain task-related labels. Recent advances in self-supervised learning (SSL) can provide means to benefit from such data in an unsupervised fashion, which have not yet been incorporated for differential diagnosis. A challenge for applying common SSL methods like SimCLR (Chen et al., 2020) or SwAV (Caron et al., 2020) to 3D brain MRI data is the need for large batch sizes and hence GPU memory, as they rely on hard negative samples to avoid collapse. Barlow Twins (Zbontar et al., 2021) is an alternative that eliminates the need for negative samples and naturally avoids collapse by redundancy reduction. As a result, it demonstrates better robustness to small batch sizes, which makes it well-suited for SSL in neuroimaging.

We introduce *Triplet Training* for differential diagnosis with limited target data. Triplet Training, see Figure 1, combines three learning strategies to include all relevant MRI data in training. First, self-supervision trains the network on task un-related data without target labels (UK Biobank). Second, we apply self-distillation on a task-related dataset that is created by merging data from ADNI and NIFD. Third, we fine-tune the model on a training set of the small in-house clinical data. Our results demonstrate that Triplet Training outperforms competing methods while being robust to a variety of properties.

To summarize, our key contributions are:
- Triplet Training for learning DNNs with limited target data.

- Adapting Barlow Twins as an efficient SSL algorithm on volumetric brain MRI data.
- Self-distillation to distill knowledge from the SSL-trained teacher network in combination with task-related labels.
- Reporting of test accuracy for differential diagnosis of AD and FTD on a well-characterized single-site clinical dataset.

## 1.1. Related Work

**Differential Diagnosis of AD and FTD with DNNs.** One line of research for differential diagnosis performs brain segmentation (Ma et al., 2020; Nguyen et al., 2022) and uses volume and thickness measurements for the classification. Such an approach may restrict learning general dementia-specific features across the entire brain. Motivated by the success of using a 3D-ResNet50 encoder-decoder on MRI (Hu et al., 2021) to extract latent representations for classification, we selected a 3D-ResNet as the backbone for our work.

As no public dataset exists comprising both AD and FTD patients, these methods combined ADNI and NIFD. The fundamental problem of such an approach is that datasets coincide with diagnosis; hence, it cannot be determined whether the network inadvertently learns to differentiate datasets instead of pathology (Geirhos et al., 2020). Thus, we incorporate ADNI and NIFD in Triplet Training for pretraining and evaluate on the in-house single-site dataset to allow for a reliable performance assessment.

**Self-Supervised Learning and Self-Distillation in Medical Image Analysis.** A variety of research (Azizi et al., 2021; Chaitanya et al., 2020; Chen et al., 2019; Taleb et al., 2020; Zhou et al., 2020; Hosseinzadeh Taher et al., 2021; Li et al., 2021; Tran et al., 2022; Zhou et al., 2019) concluded that self-supervised pre-training on domain-related datasets (i.e., unlabeled (3D) medical images) improves performance on medical downstream tasks. Haghighi et al. (2022) added restorative and adversarial branches to the SSL pipeline for medical downstream tasks. Additionally, Jiang and Miao (2022) and Ye et al. (2022) showed how SSL trained on task-unrelated medical images improves generalization on low-data regimes. This problem has also been tackled with self-distillation in Paluru et al. (2023), Li et al. (2022), and Sun et al. (2021).

In summary, self-supervised pre-training and self-distillation on medical images improve the performance of the downstream task, with domain-related datasets adding additional benefits. Such approaches have not yet been explored for differential diagnosis and have not yet been extended to Triplet Training. Moreover, research on Barlow Twins has been limited despite its attractive properties for volumetric medical images.

## 2. Methods

In this section, we present the details of Triplet Training to tackle the limited data availability for the target task. We utilize SSL with Barlow Twins to integrate task un-related data in the initial step. In the second step, we propose to include task-related data via self-distillation. Self-distillation fully utilizes the previous SSL step by aligning the distribution of latent features extracted by the student network with those learned from SSL, using the Kullback-Leibler (KL) divergence. This method not only builds on prior learning but also reduces the risk of overfitting on the task-related dataset. Finally, we fine-tune the

Table 1: Statistics for unlabeled $\mathcal{U}$, task-related $\mathcal{D}$, and target $\mathcal{T}$ datasets. MMSE denotes the Mini Mental State Examination score.

| Dataset | Diagnosis | # Samples | % Female | Age | MMSE |
|---|---|---|---|---|---|
| $\mathcal{U}$ = UK Biobank | N/A | 39,560 | 52.6 | $63.6 \pm 7.5$ | N/A |
| $\mathcal{D}$ = ADNI+NIFD | CN | 766 | 56.9 | $71.9 \pm 7.1$ | $29.0 \pm 1.2$ |
| | AD | 489 | 44.2 | $74.4 \pm 7.7$ | $22.0 \pm 4.1$ |
| | FTD | 50 | 28.0 | $60.8 \pm 6.3$ | $24.1 \pm 5.8$ |
| $\mathcal{T}$ = In-House | CN | 143 | 46.9 | $64.2 \pm 9.9$ | N/A |
| | AD | 110 | 50.0 | $67.3 \pm 8.4$ | N/A |
| | FTD | 76 | 50.0 | $64.6 \pm 9.4$ | N/A |

model on the target dataset. Before going into technical details, we introduce notation and datasets.

## 2.1. Preliminaries and Datasets

We define a 3D image as $\mathcal{I} \in \mathbb{R}^{H \times W \times D}$, with $H$, $W$, $D$ as height, width and depth, respectively. A dataset consists of $N$ 3D images $\mathcal{I}_i$, $i = 1, \ldots, N$, and class labels $y_i$ if available. Our model consists of a feature extractor $f : \mathbb{R}^{H \times W \times D} \to \mathbb{R}^Z$, with $Z$ the latent space dimension, and a projection head $g : \mathbb{R}^Z \to \mathbb{R}^C$, which maps the latent vectors to outputs of dimension $C$. We select a 3D-ResNet backbone for the feature extractor $f$ and a two-layer MLP for the projection head $g$ (implementation details in Section A).

We utilize three datasets:

1. The unlabeled dataset $\mathcal{U}$ comprises $N = 39,560$ samples $X_i^{\mathcal{U}} = (\mathcal{I}_i^{\mathcal{U}})$ extracted from the UK Biobank (Miller et al., 2016).

2. The labeled, task-related dataset $\mathcal{D}$ consists of $N = 1,305$ samples $X_i^{\mathcal{D}} = (\mathcal{I}_i^{\mathcal{D}}, y_i^{\mathcal{D}}), y_i^{\mathcal{D}} \in \{CN, AD, FTD\}$ from ADNI and NIFD.

3. The labeled target in-house dataset $\mathcal{T}$ consists of $N = 329$ samples $X_i^{\mathcal{T}} = (\mathcal{I}_i^{\mathcal{T}}, y_i^{\mathcal{T}})$, $y_i^{\mathcal{T}} \in \{CN, AD, FTD\}$ from hospital Klinikum rechts der Isar, Munich, Germany.

Table 1 reports demographic statistics for all three datasets.

## 2.2. Triplet Training

**1. Self-Supervised Learning.** The self-supervision task proposed in Barlow Twins (BT) de-correlates features in latent space and has shown to be relatively robust with respect to the batch size (Zbontar et al., 2021). This benefits training with 3D medical images because their large size limits batch sizes. Hence, BT presents a promising approach for the initial step of Triplet Training.

To pre-train the feature extractor $f^\theta$ with trainable parameters $\theta$ on the unlabeled dataset $\mathcal{U}$, two different augmentations $A$ and $B$ of an input image $\mathcal{I}_i^{\mathcal{U}}$ are required. These augmented images $A(\mathcal{I}_i^{\mathcal{U}})$ and $B(\mathcal{I}_i^{\mathcal{U}})$ are fed into a neural network consisting of the feature extractor $f^\theta$ and a projection head $g^\theta$, yielding two output latent vectors $z_i^A = g^\theta(f^\theta(A(\mathcal{I}_i^{\mathcal{U}})))$ and $z_i^B = g^\theta(f^\theta(B(\mathcal{I}_i^{\mathcal{U}}))), z_i^A, z_i^B \in \mathbb{R}^C$. The model is optimized by maximizing the cross-correlation between corresponding features of different augmentations

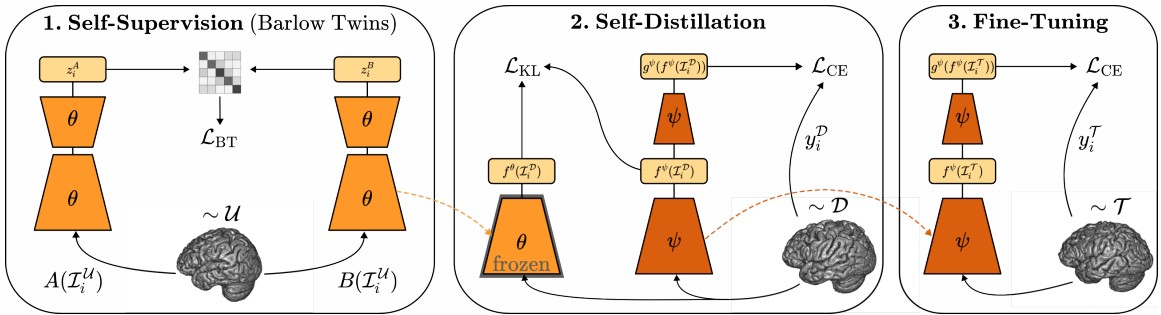

Figure 2: Overview of the three stages of Triplet Training.

$\mathcal{C}_{cc}$ and minimizing the cross-correlation between the remaining components $\mathcal{C}_{cj}$:

$$\mathcal{L}_{\text{BT}} = \sum_c (1 - \mathcal{C}_{cc})^2 + \lambda_1 \sum_c \sum_{j \neq c} \mathcal{C}_{cj}{}^2, \ \text{ with } \ \mathcal{C}_{cj} = \frac{\sum_i z_{i,c}^A z_{i,j}^B}{\sqrt{\sum_i (z_{i,c}^A)^2}\sqrt{\sum_i (z_{i,j}^B)^2}}$$

with $c = 1, \ldots, C$ indices across the latent space dimension $C$, $i$ the index of a sample within the dataset $\mathcal{U}$, and $\lambda_1$ a constant hyper-parameter. This loss makes embeddings invariant to distortions while also reducing redundant information. We denote the resulting weights after this self-supervised pre-training step as $\theta'$.

**2. Self-Distillation.** This step requires the feature extractor $f^\theta$, with pre-trained weights $\theta = \theta'$ from the previous step, as a teacher. We freeze the teacher network $f^\theta$ during training to reduce the risk of over-fitting towards the task-related dataset $\mathcal{D}$. We randomly initialize a student network $f^\psi$ with trainable parameters $\psi$ of the same architecture as the teacher, and an additional projection head $g^\psi$. Inspired by Tian et al. (2020), the student is trained on the task-related dataset $\mathcal{D}$ by minimizing the KL divergence $\mathcal{L}_{\text{KL}}$ between the outputs of the feature extractors $f^\theta(\mathcal{I}_i^\mathcal{D})$ and $f^\psi(\mathcal{I}_i^\mathcal{D})$, and minimizing the cross-entropy $\mathcal{L}_{\text{CE}}$ between the predictions of the student $g^\psi(f^\psi(\mathcal{I}_i^\mathcal{D}))$ and the related class labels $y_i^\mathcal{D}$:

$$\mathcal{L}_{\text{SD}} = \lambda_2 \mathcal{L}_{\text{KL}}(\mathcal{Z}^\psi \sim f^\psi(\mathcal{I}_i^\mathcal{D}), \mathcal{Z}^\theta \sim f^\theta(\mathcal{I}_i^\mathcal{D})) + (1 - \lambda_2) \sum_i \mathcal{L}_{\text{CE}}(g^\psi(f^\psi(\mathcal{I}_i^\mathcal{D})), y_i^\mathcal{D}),$$

with $\mathcal{Z}^\theta$ and $\mathcal{Z}^\psi$ random variables sampled via forward passes of the samples from the dataset $\mathcal{D}$, and $\lambda_2$ a constant hyper-parameter trading off the importance of the first and second terms of $\mathcal{L}_{\text{SD}}$. The resulting weights of the student network are denoted as $\psi'$.

**3. Fine-Tuning.** In the final step, we optimize the student network $f^\psi$, $g^\psi$ initialized with pre-trained weights $\psi = \psi'$ from the previous step, by fine-tuning it on the in-house dataset $\mathcal{T}$ for the target task using cross-entropy loss:

$$\mathcal{L}_{\text{FT}} = \sum_i \mathcal{L}_{\text{CE}}(g^\psi(f^\psi(\mathcal{I}_i^\mathcal{T})), y_i^\mathcal{T}).$$

## 3. Experiments

**Pre-processing and Data Augmentation:** Each T1-weighted MRI scan is pre-processed using SPM[1] and the VBM pipeline of CAT12 (Gaser et al., 2022). The results are gray-matter density volumes (samples with a quality control score lower than B– are discarded),

---

1. https://www.fil.ion.ucl.ac.uk/spm/software/spm12

Table 2: Mean, standard deviation, and pairwise p-values of the balanced accuracy (BAcc), true positive rate per class (TPR), and macro-F1 score (F1) across splits for 3-class differential diagnosis.

| Training Strategy | $\mathcal{U}$ | $\mathcal{D}$ | $\mathcal{T}$ | $\mathrm{BAcc}_{\mathcal{T}}$ | $p$-value | $\mathrm{TPR_{CN}}$ | $\mathrm{TPR_{AD}}$ | $\mathrm{TPR_{FTD}}$ | $\mathrm{F1}_{\mathcal{T}}$ | $\mathrm{BAcc}_{\mathcal{D}}$ |
|---|---|---|---|---|---|---|---|---|---|---|
| Supervised | | | ✓ | $67.15 \pm 5.36$ | 0.011 | 69.9 | 65.5 | 65.8 | $66.94 \pm 5.52$ | - |
| Supervised | | ✓ | ✓ | $68.44 \pm 4.63$ | 0.016 | 79.7 | 66.4 | 59.2 | $69.78 \pm 4.26$ | 78.2 |
| Self-Supervised (SimCLR) (Chen et al., 2020) | ✓ | | ✓ | $63.47 \pm 4.38$ | 0.001 | **86.0** | 50.0 | 54.0 | $64.44 \pm 4.13$ | - |
| Self-Supervised (VICReg) (Bardes et al., 2022) | ✓ | | ✓ | $68.94 \pm 3.42$ | 0.012 | 72.7 | 70.0 | 64.5 | $69.22 \pm 2.78$ | - |
| Self-Supervised (DiRA) (Haghighi et al., 2022) | ✓ | | ✓ | $66.78 \pm 0.89$ | 0.001 | 80.4 | 60.9 | 59.2 | $67.21 \pm 2.03$ | - |
| Self-Supervised (BT) (Zbontar et al., 2021) | ✓ | | ✓ | $71.36 \pm 4.18$ | 0.072 | 79.7 | 68.2 | 65.8 | $72.24 \pm 3.78$ | - |
| Triplet Training (Ours) | ✓ | ✓ | ✓ | **75.57** $\pm$ 3.62 | - | 81.8 | **71.8** | **73.7** | **75.32** $\pm$ 4.51 | **85.6** |

which are min-max rescaled, center-cropped, and resampled to a spatial dimension of $55 \times 55 \times 55$ (for training convenience without sacrificing model performance). Section B.2 reports details about the data augmentation strategy.

**Evaluation:** As the target dataset $\mathcal{T}$ is relatively small, we perform 5-fold cross-validation with ratios of 65%, 15%, and 20% for train, validation, and test sets, respectively, stratified by age, gender, and diagnostic labels to prevent biased results (Barnes et al., 2010). Additionally, we split a balanced 20%-portion of the task-related dataset $\mathcal{D}$ to perform further evaluations for the task at hand.

**Miscellaneous:** Hyper-parameters for the individual training steps and search spaces of baseline methods are reported in Section B.1. We implement models with PyTorch (Paszke et al., 2019) and train on one NVIDIA GeForce 3090 with 24 GByte memory. We train the model for 29,300 self-supervised iterations (24 hours), followed by 600 self-distillation iterations (2.5 hours) and 150 fine-tuning iterations with early stopping (40 minutes).

## 4. Results

As a baseline, we implement a non-deep learning approach for the differential diagnosis on $\mathcal{T}$, by extracting FreeSurfer (Fischl, 2012) volume and thickness features from MRI scans to train an XGBoost classifier, which achieves a balanced accuracy (BAcc) of $66.46 \pm 3.45\%$.

As seen in Table 2, training a DNN on the target dataset $\mathcal{T}$ alone results in a BAcc of $67.15 \pm 4.78\%$, which is likely due to the overfitting on the small task-specific data. Pre-training the model on the task-related dataset $\mathcal{D}$ improves the performance only marginally by 1.29%. Pre-training with unlabeled $\mathcal{U}$ with established SSL methods (SimCLR (Chen et al., 2020), VICReg (Bardes et al., 2022), DiRA (Haghighi et al., 2022), and Barlow Twins (Zbontar et al., 2021)) and then fine-tuning on $\mathcal{T}$ outperforms supervised pre-training on $\mathcal{D}$ by 2.92% (with Barlow Twins). Triplet Training, which adds a self-distillation step on $\mathcal{D}$ after self-supervised pre-training, significantly outperforms all competing approaches on the target dataset, achieving a BAcc of $75.57 \pm 3.62\%$ with the highest true positive rates for both types of dementia (see Table 2).

Additionally, we evaluate Triplet Training on the hold-out test set of $\mathcal{D}$ after self-distillation on $\mathcal{D}$, which clearly outperforms (+7.4%) supervised training on $\mathcal{D}$ alone (de-

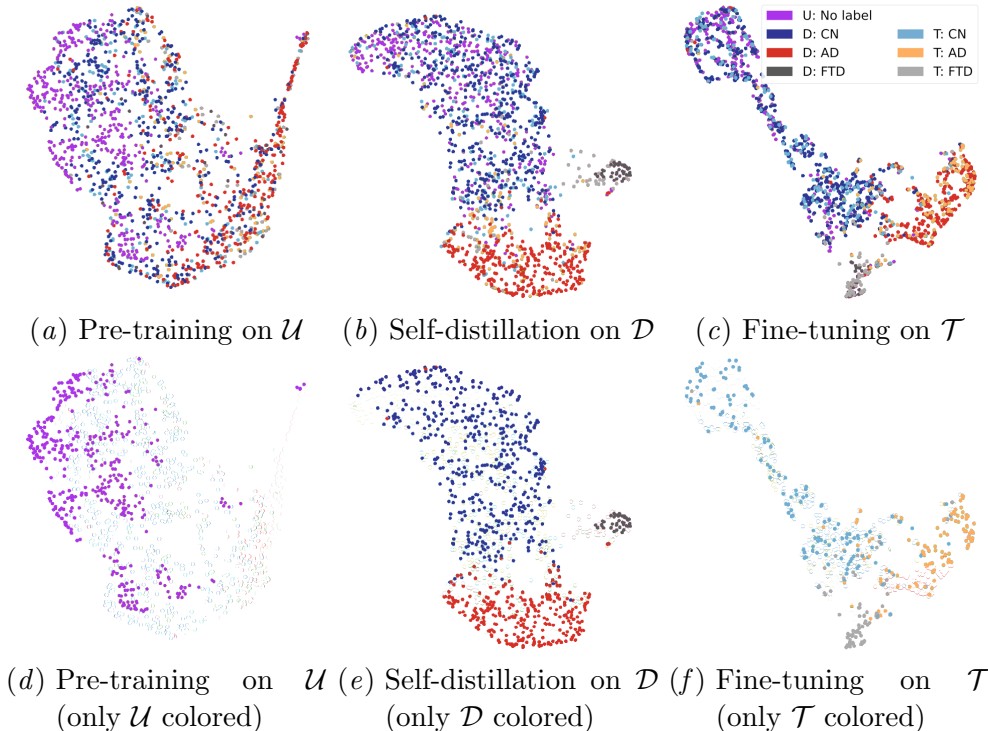

($a$) Pre-training on $\mathcal{U}$     ($b$) Self-distillation on $\mathcal{D}$     ($c$) Fine-tuning on $\mathcal{T}$

($d$) Pre-training    on    $\mathcal{U}$     ($e$) Self-distillation on $\mathcal{D}$     ($f$) Fine-tuning    on    $\mathcal{T}$
(only $\mathcal{U}$ colored)            (only $\mathcal{D}$ colored)            (only $\mathcal{T}$ colored)

Figure 3: Changes in latent space of all datasets (first row) and the step-wise target dataset (second row) after each step in Triplet Training with UMAP. $\mathcal{U}$: No label (purple, representative fraction of samples to improve readability); Task-related $\mathcal{D}$: CN (dark blue), AD (red), FTD (dark grey); In-house $\mathcal{T}$: CN (light blue), AD (orange), FTD (light grey).

noted as $\text{BAcc}_{\mathcal{D}}$ in Table 2). This indicates that Triplet Training potentially mitigates overfitting when training with limited data, thus, extracts features that generalize well.

**Visualization of the latent space.** We argue that the high accuracy of Triplet Training is rooted in decision boundaries of the classifier that are less population dependent. Therefore, we plot the evolution of the latent features of all three datasets $\mathcal{U}$, $\mathcal{D}$ and $\mathcal{T}$ after each step in Triplet Training with UMAP (McInnes et al., 2018), visualized in Figure 3. After self-supervised pre-training on $\mathcal{U}$ only, all samples of different classes from the three datasets are mixed together. After self-distillation on $\mathcal{D}$, there is a trend of separation between CN, AD, and FTD samples from all datasets. The unlabeled samples drawn from $\mathcal{U}$ display considerable overlap with the CN samples, which aligns with expectations as the majority of the UK Biobank samples consist of healthy individuals. Furthermore, the final features extracted after full Triplet Training are well separated for each class without dataset dependence, with a particularly clean cluster of FTD samples from $\mathcal{D}$ and $\mathcal{T}$. Moreover, CN and AD samples of $\mathcal{D}$ maintain a clear separation, indicating that the network did not unlearn the previous knowledge while fitting on the new domain. This property is crucial in continual learning and domain adaptation, showing that Triplet Training generalizes well even with limited data available for the target task.

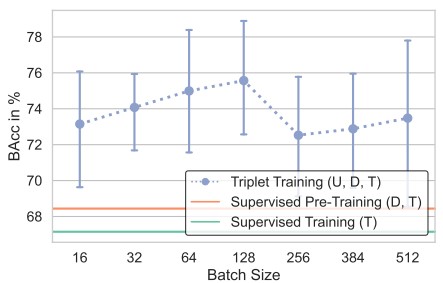
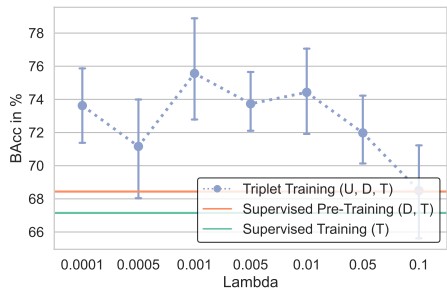

(*a*) BAcc for different batch sizes.    (*b*) BAcc for different values of $\lambda_2$.

Figure 4: Ablation studies of hyper-parameters in Triplet Training.

**Ablation Study 1: Hyper-parameters.**    As shown in the original work (Zbontar et al., 2021), Barlow Twins is relatively robust to the batch size. However, the evaluated batch sizes up to 4,096 are infeasible when working with volumetric images. Thus, we examine the robustness of Triplet Training w.r.t. batch sizes typically used in DNNs for medical image analysis. As seen in Figure 4(*a*), Triplet Training consistently surpasses both supervised training on $\mathcal{T}$ and pre-training on $\mathcal{D}, \mathcal{T}$ across all batch sizes, with 128 (used for all experiments) achieving the highest performance marginally over the other batch sizes. Evidently, Triplet Training benefits from a moderate increase in batch size and surpasses all competing methods regardless of the batch size, demonstrating considerable robustness to the batch size variation. Figure 4(*b*) shows that Triplet Training outperforms the baseline methods for a wide range of $\lambda_2$, a constant hyper-parameter used during self-distillation.

**Ablation Study 2: Benchmark Self-Supervised Approaches.**    We replace the SSL algorithm (Barlow Twins) in the initial step of Triplet Training with three SOTA algorithms. Table 3 reports that Triplet Training showcases high and consistent accuracy across all SSL methods, highlighting its robustness and generalizability. Among them, Barlow Twins and SimCLR demonstrate the best performance, and introduce few additional hyper-parameters compared to the other methods. We argue that Barlow Twins is the optimal choice, as it has shown to be robust in terms of the batch sizes.

Table 3: Mean and standard deviation of the balanced accuracy (BAcc), true positive rate (TPR), and macro-F1 score (F1) for different SSL approaches in the initial step of the Triplet Training. We propose to use Barlow Twins (BT) in Triplet Training.

| SSL in Triplet Training | $BAcc_{\mathcal{T}}$ | $TPR_{CN}$ | $TPR_{AD}$ | $TPR_{FTD}$ | $F1_{\mathcal{T}}$ | $BAcc_{\mathcal{D}}$ |
|---|---|---|---|---|---|---|
| SimCLR (Chen et al., 2020) | $75.22 \pm 2.80$ | **86.7** | 69.1 | 69.7 | **$75.64 \pm 2.74$** | **86.0** |
| VicReg (Bardes et al., 2022) | $73.44 \pm 4.92$ | 83.9 | 69.1 | 67.1 | $74.15 \pm 4.91$ | 85.5 |
| DiRA (Haghighi et al., 2022) | $74.49 \pm 4.14$ | 86.7 | 65.5 | 71.1 | $74.85 \pm 4.03$ | 85.4 |
| BT (Zbontar et al., 2021) | **$75.57 \pm 3.62$** | 81.8 | **71.8** | **73.7** | $75.32 \pm 4.51$ | 85.6 |

## 5. Conclusion

We introduced Triplet Training for differential diagnosis of dementia, which enhances predictive performance for tasks with limited data availability. Triplet Training consists of three steps that fully utilize large-scale unlabeled data, task-related data, and limited amounts of target data, achieving a BAcc of 75.6% on a well-characterized clinical dataset while showing strong generalizability. Ablation studies confirmed Triplet Training's robustness against varying hyper-parameters and method selection in the initial step.

## Acknowledgments

This research was supported by the Federal Ministry of Education and Research in the call for Computational Life Sciences (DeepMentia, 031L0200A) and the DFG. This research was conducted using the UK Biobank Resource. The authors gratefully acknowledge the Leibniz Supercomputing Centre for funding this project by providing computing time on its Linux-Cluster.

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

## Appendix A. Architecture

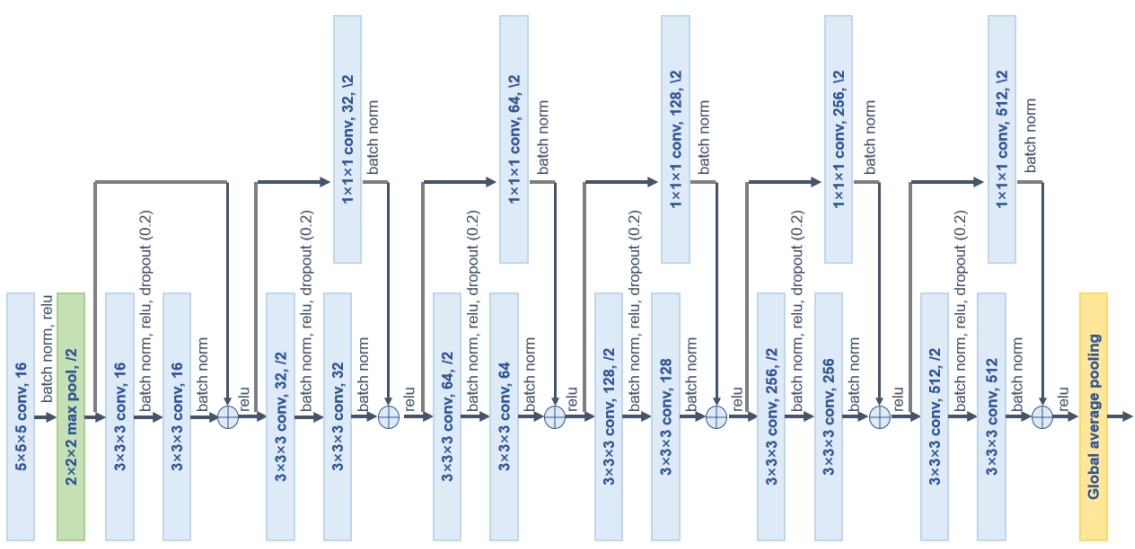

Figure 5: We select a 3D ResNet as the feature extractor $f$ for all models. It consists of six residual blocks, each consisting of two convolutional layers followed by batch normalization and ReLU non-linearity. The five last residual blocks each start with a convolutional layer with stride two.

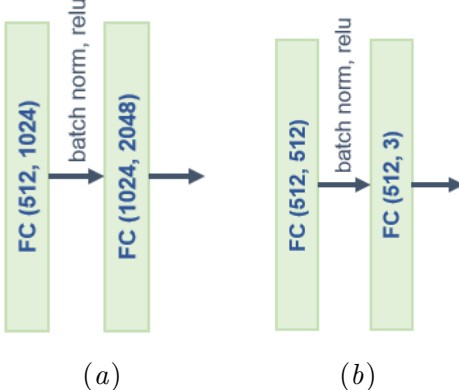

$(a)$ $(b)$

Figure 6: Projection head $g$ for: (a) self-supervision (Barlow Twins); (b) self-distillation and fine-tuning.

# Appendix B. Training Details

## B.1. Hyper-parameters

Table 4: Hyper-parameters of the different training strategies. The number of iterations for each step is based on the convergence of the validation set. If available, we use the hyper-parameters proposed in the original work.

| Training Strategy | Hyper-Parameter | Value |
|---|---|---|
| Supervised Training ($\mathcal{T}$) | Learning rate | 0.01 |
| | Weight decay | 0.00001 |
| | Batch size | 64 |
| | Training iterations | 150 |
| Supervised Pre-Training ($\mathcal{D}$) | Learning rate | 0.01 |
| | Weight decay | 0.0000015 |
| | Batch size | 128 |
| | Training iterations | 600 |
| Triplet Training (Self-Supervision) | Learning rate | 0.5 |
| | Weight decay | 0.0000015 |
| | Batch size | 128 |
| | Training iterations | 29,300 |
| | $\lambda_1$ | 0.005 |
| Triplet Training (Self-Distillation) | Learning rate | 0.01 |
| | Weight decay | 0.0000015 |
| | Batch size | 128 |
| | Training iterations | 600 |
| | $\lambda_2$ | 0.001 |
| Triplet Training (Fine-Tuning) | Learning rate | 0.0005 |
| | Weight decay | 0.00001 |
| | Batch size | 64 |
| | Training iterations | 150 |

## B.2. Data Augmentation

Table 5: Data Augmentations used in the Triplet Training.

| Training Strategy | Augmentation | Values |
|---|---|---|
| Self-Supervision | Rescale Intensity | intensity range = (0, 1) |
| | Random Cropping with Resizing | crop scale = (0.5, 1.0)
output size = (55, 55, 55)
random center = True |
| | Random Flipping | axes = (0, 1, 2)
probability = 0.5 |
| | Random Affine Transformation | rotation range = $(-90°, +90°)$
translation range = $(-8$ pixel, $+8$ pixel$)$
probability = 0.5 |
| Self-Distillation | Rescale Intensity | intensity range = (0, 1) |
| | Random Affine Transformation | rotation range = $(-8°, +8°)$
translation range = $(-8$ pixel, $+8$ pixel$)$
probability = 0.5 |
| Fine-Tuning | Rescale Intensity | intensity range = (0, 1) |
| | Random Affine Transformation | rotation range = $(-8°, +8°)$
translation range = $(-8$ pixel, $+8$ pixel$)$
probability = 0.5 |

