# OpenReview forum: "From Barlow Twins to Triplet Training: Differentiating Dementia with Limited Data"
_MIDL.io/2024/Conference — MIDL 2024 Oral_

### Official Review · Reviewer_ZGAb · 2024-02-28

**Confidence:** 5
**Preliminary Rating:** 4
**Recommendation:** Oral
**Final Rating:** 4

**Summary:**

The authors are proposing a training strategy for classification of neurodegeneration (AD vs CN vs FTD) using a DL model. They propose 3 steps. 1) train a backbone encoder on a very large dataset using a barlow twin approach. 2) train a second encoder using distillation from the first model on a smaller, but more relevant dataset. 3) fine tune the training on a small target dataset.

Results are better than straightforward training designs (although still not good enough to be clinically useful). The experiments are convincing and are well explained given the limited space available.

**Strengths:**

+ the triplet training approach makes sense
+ good approach to use gray matter density instead of actual MRI, this likely removes much of the site dependent variability (although some still likely exist and is not investigated).
+ good experimental design
+ paper is clear
+ results are encouraging

**Weaknesses:**

The premise of the paper is not correct. Structural MRI is not the primary method for the diagnosis of dementia. Dementia is commonly clinically diagnosed with neuropsy evaluation. More specific diagnosis is achieved in most cases with CSF or PET markers for pathological proteins (e.g. Tau, Amyloid,…). MRI is used mostly for screening out non degenerative causes (e.g. cancer) or to refine/confirm the diagnosis in some disorders (e.g FTD).

The rational for the paper is still correct though: that MRI can be useful for differential diagnosis (e.g. AD Vs FTD), that disorders often overlap, and that there is a lack of large databases covering most disorders.

The authors should give more consideration to the label ground truth used. Are they using the clinical ADNI label (mostly based on MMSE) or do they use a more recent label using other biomarkers (e.g. Amyloid)? This is important because the clinical diagnosis of AD for example is only 80% sensitive and 70% specific. More detail about this should be provided.

3/ a few critical details are missing (e.g. above), including what are the methods used for data augmentation of the first step for the BT strategy?

**Detailed Comments:**

The datasets used are likely using different MRI acquisition parameters and therefore it is unclear what is the impact on the latent variables between them due to site difference compared to differences due to the disorders.

Same comment about the population mix (age, gender, ApoE4,… ).

**Justification Of Final Rating:**

This is a good paper and the authors have added more information about the double Autoencoder scheme which was a question raised by most of the reviewers. The results are very encouraging, although lacking some quantitative validation.

**Justification Of The Preliminary Rating:**

Good and well written paper with an interesting approach. Some critical details are missing, and the clinical justification needs some refinement. It would be interesting to understand how the datasets differ in terms of acquisition and population.

**Questions To Address In The Rebuttal:**

Are all the datasets using similar MRI sequences?

Are the datasets different in terms of main known risk factors (age, gender, ApoE4,…)

How is data augmentation implemented for the BT learning?

**Special Issue:**

No

---

> ### Author Response · Authors · 2024-03-15
>
> We greatly appreciate your very in-depth review and insightful suggestions for the improvement of the paper. We hope the following clarifications can address your questions and concerns, and we are open to further questions:
> 1. Thanks for pointing out the detailed medical premises of structural MRI. We took the chance to improve the manuscript and rephrased the corresponding paragraphs. We target using MRI for the differential diagnosis of dementia.
>
> 2. All datasets acquire MPRAGE images (with some ADNI data for MPRAGE/IR-SPGR). We extracted volumetric gray matter density maps from T1-weighted MRI using the VBM pipeline of CAT12 [3] as the input.
>
> 3. We show the statistics of all the datasets in Table 1 of the manuscript, which includes the information of age and gender across datasets. Regrettably, we don’t have the complete information for ApoE4 for the in-house data available.
>
> 4. As harmonizing high-dimensional images across sites is still ongoing research with the risk of removing disease-related information, in our study, we use gray matter density maps as the input, an established approach for standardizing the input with respect to scanner effects [4].
>
> 5. We performed random cropping with resizing, random flipping, and random affine transformation for data augmentation in SSL training. Detailed information can be found in the Supplementary Table 5.
>
> [3] Gaser C, Dahnke R, Thompson PM, Kurth F, Luders E, Alzheimer"s Disease Neuroimaging Initiative. CAT - A Computational Anatomy Toolbox for the Analysis of Structural MRI Data. bioRxiv 2022.06.11.495736; doi: https://doi.org/10.1101/2022.06.11.495736
>
> [4] Dhinagar, N.J., Thomopoulos, S.I., Owens-Walton, C., Stripelis, D., Ambite, J.L., Steeg, G.V. and Thompson, P.M. (2022), Alzheimer’s Disease Detection with a 3D Convolutional Neural Network using Gray Matter Maps from T1-weighted Brain MRI. Alzheimer's Dement., 18: e066446. https://doi.org/10.1002/alz.066446

---

### Official Review · Reviewer_KNGw · 2024-02-29

**Confidence:** 3
**Preliminary Rating:** 4
**Final Rating:** 5

**Summary:**

The authors propose Triplet Training, a combination of representation learning approaches that has not yet been explored for differential diagnosis. The framework involves three stages: self-supervised pre-training using Barlow Twins, self-distillation on task relevant data, and fine-tuning on targeted small datasets, leading to improvements in diagnostic accuracy in classifying two types of dementia and control cases.

**Strengths:**

The paper is written in a clear and engaging way. It explains its ideas well and sets up its experiments carefully. The applicability of the proposed approach extends beyond the paper's immediate scope, interesting and highly relevant to the MIDL community. The authors are shown to be meticulous; paying attention to properly stratifying their training data, especially when there is data imbalance, which we see here with low FTD cases; focusing on issues in the field such as "shortcut learning". Furthermore, the results show notable improvements gained through Triplet Learning compared to approaches that address the task independently.

**Weaknesses:**

The authors provide background by emphasizing the challenges in distinguishing between different dementia types. Although their study only includes two subtypes. It is unclear if the framework would perform on differentiating more than two subtypes given diagnosis is challenging as symptoms overlap.

**Detailed Comments:**

Minor comment:

The second most common type of dementia is vascular [1], while FTD is the third most common [2].

[1] https://www.ncbi.nlm.nih.gov/pmc/articles/PMC6039843
[2] https://www.ncbi.nlm.nih.gov/pmc/articles/PMC5761910

**Justification Of Final Rating:**

I would like to thank the authors for adequately addressing all my questions, and the questions of the other reviewers. I've decided to increase my initial score, believe it accurately reflects the paper's quality and merits within the context of its field.

**Justification Of The Preliminary Rating:**

The methodology proposed not only serves the objectives of this study but also shows potential applicability in broad, therefor it could bring interesting discussions to the conference. The experiments align with current research trends, are carefully thought out and the manuscript is well written.

**Questions To Address In The Rebuttal:**

Could you please clarify the nature of the input provided to the neural networks? Specifically, are you utilizing volumetric gray matter density maps, and is white matter excluded from consideration?

Will the code associated with this study will be made publicly accessible upon publication?

---

> ### Author Response · Authors · 2024-03-15
>
> We greatly appreciate your positive review and detailed comments, we hope the following clarifications can address your questions and we are open to further questions:
> 1. We would gladly validate our framework in distinguishing more than two dementia subtypes once we collect enough data for the rare dementia types. Unfortunately, we are limited by the availability of publicly available datasets including more than these two subtypes.
>
> 2. The input of the neural networks is the volumetric gray matter density maps extracted via the VBM pipeline of CAT12 [3], excluding the white matter. Using gray matter density maps as the input is an established approach for standardizing the input with respect to scanner effects [4].
>
> 3. We have added the link of the code to the manuscript and we are eager to make our code public upon acceptance to facilitate reproducibility.
>
> [3] Gaser C, Dahnke R, Thompson PM, Kurth F, Luders E, Alzheimer"s Disease Neuroimaging Initiative. CAT - A Computational Anatomy Toolbox for the Analysis of Structural MRI Data. bioRxiv 2022.06.11.495736; doi: https://doi.org/10.1101/2022.06.11.495736
>
> [4] Dhinagar, N.J., Thomopoulos, S.I., Owens-Walton, C., Stripelis, D., Ambite, J.L., Steeg, G.V. and Thompson, P.M. (2022), Alzheimer’s Disease Detection with a 3D Convolutional Neural Network using Gray Matter Maps from T1-weighted Brain MRI. Alzheimer's Dement., 18: e066446. https://doi.org/10.1002/alz.066446

---

### Official Review · Reviewer_ig5f · 2024-03-03

**Confidence:** 4
**Preliminary Rating:** 5
**Recommendation:** Oral

**Summary:**

This work proposed a three-stage training architecture to address the limited data available for AI diagnosis of dementia. The first stage uses self-supervision to learn the stable features invariant through different augmentations. Then, a self-distillation network is trained to regress the task-related label from the extracted features in Stage 1. Lastly, pre-train the second regression network on a limited private dataset. From the experiments, the newly proposed architecture performs well on the private clinical dataset and shows strong generalizability.

**Strengths:**

The paper is written in a clear and engaging way. It explains its ideas well and sets up its experiments carefully. The applicability of the proposed approach extends beyond the paper's immediate scope, interesting and highly relevant to the MIDL community. The proposed architecture is novel and well-designed, which makes good sense to address the data limits. The formulations and analysis are solid， combined with the well-organized manuscript and delicate visualizations. This method presents the universal potential to expand onto variant tasks. Good Job.

**Weaknesses:**

The Ablation Experiments seem to over-focus on details like hyper-parameters and benchmarks rather than the architecture design. The ablation should contain more essential compressions, like Why self-distillation is better than the directly fine-tuning or frozen feature extractor followed by a tail regressor in stage 2.

**Detailed Comments:**

Trying to apply the same strategy to a multi-modality dataset seems interesting.

**Justification Of The Preliminary Rating:**

The methodology proposed not only serves the objectives of this study but also shows potential applicability in broad, therefor it could bring interesting discussions to the conference. The experiments align with current research trends, are carefully thought out and the manuscript is well written.

**Questions To Address In The Rebuttal:**

More relevant ablation with analysis in detail to prove the key advantage of the design.

**Special Issue:**

Yes

---

> ### Author Response · Authors · 2024-03-15
>
> We greatly appreciate your very positive review and constructive suggestions. We plan to explore the same strategy for a multi-modality dataset in future work. In this study, we focused our ablation study on validating the robustness of Triplet Training across hyper-parameters and SSL algorithms in the initial stage. We evaluated direct fine-tuning and frozen feature extractors during the early design of our pipeline, which all failed to surpass the current one (BAcc of 69.81 +- 4.52% with frozen feature extractors during fine-tuning and 72.40 +- 4.36% with direct fine-tuning on D). We argue that self-distillation marginalizes the risk of overfitting on the task-related dataset compared to the other approaches.

---

### Official Review · Reviewer_YJmH · 2024-03-04

**Confidence:** 4
**Preliminary Rating:** 4

**Summary:**

This article primarily introduces Triplet Training as a solution for the limited dataset challenge in differential diagnosis of Alzheimer's Disease (AD) and Frontotemporal Dementia (FTD). By incorporating Barlow Twins for self-supervised pre-training on unlabeled data and utilizing self-distillation on task-related data with a teacher-student network, useful knowledge is obtained. This allows leveraging abundant unlabeled data to assist model training. The method is logically rigorous, theoretically sound, and excellently written.

**Strengths:**

The overall operational procedure is clearly described, with rich novelty in the ideas presented. The illustrations are clear and easy to understand. In terms of experimentation, there is a significant improvement compared to previous baseline models. Overall, from theoretical innovation to experimental results and writing, the quality of the work is commendable.

**Weaknesses:**

It is recommended to open-source the network and provide reproducible code. Additionally, can this network be utilized for clinical diagnosis? Deployment and implementation should be considered in the future.
In the manuscript, you mentioned that the evaluation method is: "As the target dataset T is relatively small, we perform 5-fold cross-validation with ratios of 65%, 15%, and 20% for train, validation, and test sets, respectively, stratified by age, gender, and diagnostic labels to prevent biased results." You need to ensure that this ratio is applied consistently across all datasets when comparing the network with other baseline models.
Additionally, using light grey color might not be easily distinguishable; changing it to green could be a viable alternative.

**Detailed Comments:**

Open Source Code Please.

**Justification Of The Preliminary Rating:**

The overall operational procedure is clearly described, with rich novelty in the ideas presented. The illustrations are clear and easy to understand. In terms of experimentation, there is a significant improvement compared to previous baseline models. Overall, from theoretical innovation to experimental results and writing, the quality of the work is commendable.

The theoretical richness of this study is valuable, particularly in the current context of insufficient diagnostic data for dementia in medical datasets.

**Questions To Address In The Rebuttal:**

I recommend acceptance.

---

> ### Author Response · Authors · 2024-03-15
>
> We greatly appreciate your positive and detailed review, we have added the link of our code to the manuscript and we are eager to make our code public upon acceptance. We hope the following clarifications can address your questions:
> 1. The same splits of target dataset T are used to evaluate all baseline models as well as our pipeline. We adapt the data split method from ClinicaDL [2] to evenly distribute age, gender, and diagnostic labels across training, validation, and test data splits.
>
> 2. Thank you for your valuable feedback regarding the color choices in our plots. We aimed to design our plots with accessibility and clarity in mind, ensuring they are colorblind-friendly and easily interpretable. To achieve this, we carefully tested various color palettes, ultimately selecting the current scheme as it best met these inclusivity and readability standards.
>
> [2] Thibeau-Sutre, E., Díaz, M., Hassanaly, R., Routier, A., Dormont, D., Colliot, O., Burgos, N.: ‘ClinicaDL: an open-source deep learning software for reproducible neuroimaging processing‘, 2021. hal-03351976

---

### Official Review · Reviewer_Eax3 · 2024-03-04

**Confidence:** 4
**Preliminary Rating:** 5
**Recommendation:** Oral
**Final Rating:** 5

**Summary:**

The authors propose a three-stage training approach for the case of limited datasets, and present its performance for differentiating dementia from 3D MRI volumes. After a thorough description of their approach, they evaluate their model and compare it to other methods in the field, showing robustness, and the usefulness of all three stages.

**Strengths:**

Their proposed 'Triplet training' is described in detail, and the paper overall is well-written. The application of the training method for the classification of the images is also an important field, and draws attention to the lack of public data, and the limitations of training on privately collected images.

**Weaknesses:**

The description of the performed experiments, and the statistical analysis of the results could be improved to provide an even more confident argument about the performance of the proposed training approach.

**Detailed Comments:**

- The reviewer thinks a public code repository would greatly benefit the impact of the presented work.

**Justification Of Final Rating:**

I would like to thank the authors for adequately addressing all my questions, and the questions of the other reviewers. I believe the review process even further improved the quality of the manuscript, and it is suitable for acceptance.

**Justification Of The Preliminary Rating:**

The authors tackle a significant issue, and their implementation is also in a very important field. The ablation studies show that all introduced stages of the training are important. The reviewer thinks a few minor clarifications are needed to make the presented results even more convinving.

**Questions To Address In The Rebuttal:**

- Could the authors address how the very specific number of iterations have been assessed for each training stage? Was it empirical based on the validation dataset?
- Could the authors describe UMAP which was used for visualizing the latent space? Or provide a reference for it?
- The standard deviations of the results (Table 3) suggest that the performance of the four methods are similar. Did the authors perform a statistical test of significance to find BT the best model for BAcc?

**Special Issue:**

Yes

---

> ### Author Response · Authors · 2024-03-15
>
> We greatly appreciate your very positive review and suggestions to improve the manuscript, we have added the link of our code to the manuscript and we are eager to make the code public upon acceptance. We hope the following clarifications can address your questions and are open to further questions:
> 1. We determined the number of iterations for each step based on the convergence of the validation set. In addition, we also used early stopping in the last training stage to prevent overfitting. We have updated these experimental details in the manuscript.
>
> 2. Uniform Manifold Approximation and Projection (UMAP) is a dimension reduction technique that can be used for high-dimensional data visualization, similarly to t-SNE, but also for general non-linear dimension reduction. The details for the underlying mathematics can be found in reference [1], which we have also added to our manuscript.
>
> 3. The performance of the four SSL algorithms used in the initial stage of Triplet Training is consistent, suggesting that the high performance was gained through the whole designated pipeline of Triplet Training rather than the choice of SSL algorithms. This highlights the robustness and generalizability of Triplet Training. We argue that Barlow Twins is the optimal choice, as it introduces fewer additional hyper-parameters and has shown to be robust in terms of batch sizes.
>
> [1] McInnes, L, Healy, J, UMAP: Uniform Manifold Approximation and Projection for Dimension Reduction, ArXiv e-prints 1802.03426, 2018

---

### Meta-Review · Area_Chair_tjqS · 2024-04-02

**Recommendation:** Accept (Oral)
**Confidence:** 4

**Metareview:**

All reviewers found the proposed method to be novel and the results promising.
2 weak accept and 2 strong accept recommendations from reviewers.

---

### Decision · Program_Chairs · 2024-04-05

Accept (Oral)